# GAR: Carbon-Aware Routing for LLM Inference via Constrained Optimization

## Abstract

The growing deployment of large language models (LLMs) makes per-request routing essential for balancing response quality and computational cost across heterogeneous model pools. Current routing methods rarely consider sustainable energy use and $CO_2$ emissions as optimization objectives, despite grid carbon intensity varying by time and region, and models differing significantly in energy consumption. To address this gap, we introduce Green-Aware Routing (GAR), a constrained multi-objective optimization framework that minimizes per-request $CO_2$ emissions subject to explicit accuracy floors and p95-latency service-level objectives (SLOs). GAR employs adaptive constraint optimization through per-dataset floor tuning and incorporates lightweight estimators for correctness, tail latency, and carbon emissions, enabling real-time routing decisions without additional inference passes. We present GAR-PD, an online primal-dual algorithm with $O(\sqrt{T})$ regret bounds alongside practical heuristic variants (GAR-Fixed, GAR-$\varepsilon$, GAR-Target) that achieve high feasibility coverage while limiting accuracy degradation. Comprehensive experiments across standard NLP benchmarks with heterogeneous LLM pools (7B–70B) demonstrate that GAR achieves substantial carbon reductions while maintaining competitive accuracy and p95 latency guarantees, providing a practical, theoretically grounded approach to sustainable LLM inference.

## 1 Introduction

Large language models (LLMs) serving at scale increasingly rely on routing architectures that balance response quality, tail latency, and computational cost by directing queries to appropriately sized models within heterogeneous pools. Recent routing frameworks address this problem by prioritizing accuracy and latency optimization (Ong et al., 2025; Feng et al., 2025; Ding et al., 2024; Chen et al., 2023), while using monetary cost as a secondary factor once performance requirements are met. Importantly, monetary cost optimization diverges significantly from carbon minimization, as cloud pricing reflects commercial strategies rather than real-time environmental impact. However, environmental factors, specifically the energy consumption and carbon dioxide ($CO_2$) emissions of each inference request, remain notably absent from routing decision processes, handled solely via subsequent measurement and reporting (Strubell et al., 2019; Luccioni et al., 2023). This gap has significant environmental implications, as grid carbon intensity exhibits up to $10\times$ variation across regions and time periods due to renewable energy availability and fossil fuel dependencies (Dodge et al., 2022; Kaack et al., 2022). Simultaneously, model energy consumption varies with architectural choices, parameter scaling, and token generation patterns, enabling possibilities for intelligent carbon-aware routing that current systems fail to exploit (Henderson et al., 2020; Wu et al., 2022). A router optimized only for cost or latency can systematically direct traffic toward high-carbon periods or carbon-intensive regions, even when multiple models achieve equivalent accuracy. Since per-request model selection already occurs within routing infrastructure, routing represents the natural venue to treat carbon as a primary optimization objective rather than an afterthought.

Recent studies in routing systems has made significant progress toward optimizing cost-performance for efficient LLM serving (Chen et al., 2023; Ding et al., 2024; Feng et al., 2025; Ong et al., 2025) and preliminary explorations of environmental considerations (Wheeler et al., 2025). These works provide essential groundwork for quality-aware routing policies and demonstrate the potential for

incorporating sustainability objectives into model selection. However, current methodologies predominantly concentrate on post-hoc trade-offs between competing objectives or datacenter-level carbon management (Guo et al., 2023; Xu et al., 2025), while per-request routing decisions typically lack formal service-level guarantees or real-time emissions optimization integrated with time-varying grid conditions. This limitation necessitates extending routing frameworks toward principled bounded optimization that treats carbon as a first-class objective while providing rigorous quality and latency guarantees.

To address these challenges, we introduce Green-Aware Routing (GAR), a constrained optimization framework that treats carbon emissions as a first-class routing objective. GAR operates by formalizing per-request $CO_2$ minimization as a constrained multi-objective optimization problem, where routing decisions must satisfy explicit accuracy floors and p95-latency service-level objectives while minimizing environmental impact. The framework employs lightweight estimators for response quality, tail latency, and carbon emissions, enabling real-time routing decisions using monotonic energy models and time-varying grid carbon intensity data. Unlike existing approaches that optimize cost or performance in isolation, GAR provides principled constraint satisfaction through adaptive floor tuning that achieves high feasibility coverage while limiting accuracy degradation to minimal levels. We present GAR-PD, an online primal-dual algorithm with $O(\sqrt{T})$ regret bounds for rolling carbon budgets, alongside practical heuristics (GAR-Fixed, GAR-$\varepsilon$, GAR-Target) that enable flexible deployment across different carbon-quality trade-off preferences. Our evaluation demonstrates substantial emission reductions while maintaining service guarantees, with comprehensive robustness analysis under prediction miscalibration and distribution shift.

In summary, our main contributions are as follows:

- We formalize carbon-aware routing as a constrained multi-objective optimization problem, providing the first principled framework for per-request $CO_2$ minimization in LLM serving with formal quality guarantees.
- We develop GAR-PD, an online algorithm with theoretical regret bounds for carbon-constrained routing, alongside practical variants for diverse deployment scenarios.
- We establish adaptive constraint optimization techniques that achieve high feasibility coverage while maintaining service quality, enabling substantial emission reductions without compromising user experience.
- We provide comprehensive experimental validation across standard benchmarks with systematic robustness analysis, demonstrating that GAR achieves substantial improvements in service reliability while delivering significant $CO_2$ reductions.

## 2 RELATED WORK

Recent advances in LLM routing have focused on optimizing cost-performance trade-offs through diverse strategies. FrugalGPT (Chen et al., 2023) pioneered cascading approaches that sequentially query models until quality thresholds are met, while RouteLLM (Ong et al., 2025) leverages preference data from Chatbot Arena to train routing models achieving substantial cost reductions. Hybrid-LLM (Ding et al., 2024) combines synthetic preference labels with BERT-based classification, GraphRouter (Feng et al., 2025) employs heterogeneous graph networks for query-model compatibility, and RouterBench (Hu et al., 2024) provides comprehensive evaluation frameworks. More recent works explore advanced routing capabilities with CARGO (Barrak et al., 2025) introducing confidence-aware routing through embedding-based regressors and binary classifiers achieving 76.4% routing accuracy, and MasRouter (Yue et al., 2025) providing unified routing for multi-agent systems through three-layer decision architectures. Despite these advances, existing approaches optimize monetary costs or performance metrics without considering environmental factors or providing formal service-level guarantees.

Carbon-aware computing research has gained significant attention in ML systems, with frameworks like EcoServe (Li et al., 2025) achieving substantial carbon reduction through resource rightsizing and hardware recycling, Clover (Li et al., 2023) balancing performance and carbon emissions in ML inference services, and GREEN (Xu et al., 2025) providing carbon-efficient scheduling for ML jobs. For LLM-specific environmental optimization, Wheeler et al. (Wheeler et al., 2025) introduced environmentally aware routing through chain-based decomposition achieving environmental

cost reduction by routing different reasoning stages to appropriately sized models, and SPROUT (Li et al., 2024) reduces carbon footprint through generation directives rather than model routing. However, existing carbon-aware frameworks primarily focus on datacenter-level resource management or static optimization strategies, lacking real-time, per-request carbon minimization with heterogeneous model pools and formal constraint guarantees.

# 3 GAR: GREEN AWARE ROUTING FOR LLM INFERENCE

## 3.1 PROBLEM FORMULATION

We consider a pool of LLMs $\mathcal{M} = \{m_1, \ldots, m_K\}$ serving queries with carbon-aware routing. For query $x$ arriving at time $t$, each model $m \in \mathcal{M}$ induces unobserved correctness $Y_m(x) \in \{0, 1\}$, end-to-end latency $L_m(x) \in \mathbb{R}_+$, and emissions $C_m(x) \in \mathbb{R}_+$. Before inference, the router obtains lightweight predictions:

$$\hat{p}_m(x) \approx \Pr[Y_m(x) = 1], \qquad \hat{\ell}_{m,\mathrm{p95}}(x) \approx \mathrm{p95}(L_m(x)), \qquad \hat{c}_m(x, t) \approx \mathbb{E}[C_m(x, t)], \quad (1)$$

where carbon estimates integrate real-time grid intensity as $\hat{c}_m(x, t) = \mathrm{energy}(m, \mathrm{tokens}(x, m)) \times \mathrm{grid\_intensity}(t, \mathrm{region}(m))$. For robustness, we optionally apply safety margins:

$$\tilde{c}_m(x, t) = (1 + \gamma_c)\hat{c}_m(x, t), \qquad \tilde{\ell}_{m,\mathrm{p95}}(x) = (1 + \gamma_\ell)\hat{\ell}_{m,\mathrm{p95}}(x). \quad (2)$$

1) **Service-Level Constraints:** Deployments specify per-dataset accuracy floor $\tau_{d(x)} \in [0, 1]$ and p95-latency target $L$. The feasible set for request $x$ at time $t$ encompasses all models meeting these requirements:

$$\mathcal{F}(x, t) = \{ m \in \mathcal{M} : \hat{p}_m(x) \geq \tau_{d(x)}, \tilde{\ell}_{m,\mathrm{p95}}(x) \leq L \}. \quad (3)$$

2) **Rolling Carbon Budget:** For streaming workloads, we maintain a sliding ledger of realized emissions over the last $W$ requests, ensuring the window average stays within per-request budget $B$:

$$\bar{C}_{t,W} = \frac{1}{W} \sum_{i=\max(1, t-W+1)}^{t} C_i \leq B \quad (4)$$

($B$ grams / request, $W$ requests; for $t < W$ we average over the available prefix). This constraint prevents carbon debt accumulation while allowing short-term flexibility.

3) **Carbon-Aware Selection:** Given feasible set $\mathcal{F}(x, t)$, GAR selects the model minimizing predicted emissions:

$$R^{\mathrm{GAR}}(x, t) \in \underset{m \in \mathcal{F}(x, t)}{\arg\min} \tilde{c}_m(x, t), \quad (5)$$

with ties broken by lower latency, then higher accuracy. For GAR-PD, this connects to the dual update:

$$\lambda_{t+1} = \left[ \lambda_t + \eta \left( \sum_{i=t-W+1}^{t} C_i - BW \right) \right]_+, \qquad m_t \in \arg\min_{m \in \mathcal{F}(x, t)} (1 + \lambda_t) \tilde{c}_m(x, t), \quad (6)$$

where $[z]_+ = \max\{z, 0\}$. This formulation enables principled carbon minimization under explicit service-level constraints, bridging environmental objectives with production deployment requirements through real-time optimization.

## 3.2 GAR FRAMEWORK OVERVIEW

**Method Overview.** As shown in Figure 1, GAR first estimates the carbon emissions, accuracy, and latency for each model in the pool before making a routing decision. The framework then selects the model with the lowest predicted carbon emissions while ensuring accuracy and latency requirements are met. Unlike traditional routing that focuses only on cost and performance (Chen et al., 2023; Ong et al., 2025), GAR treats carbon reduction as a primary objective.

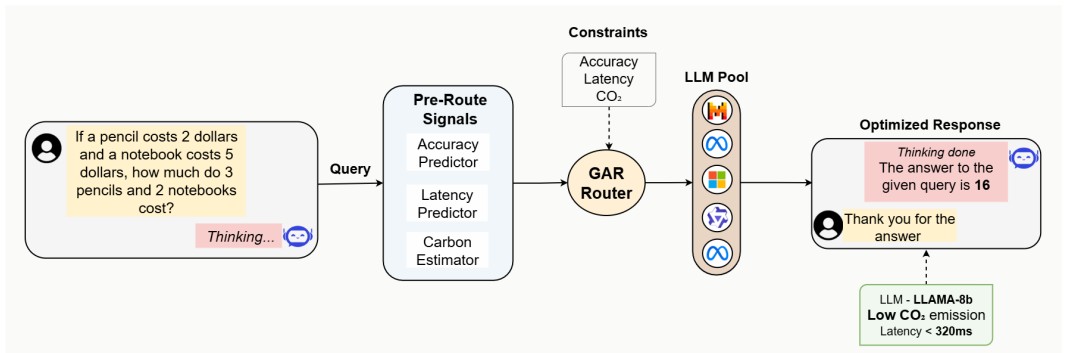

Figure 1: GAR framework

**Pre-route estimators.** GAR uses three lightweight predictors trained on historical data (Dodge et al., 2022). The quality predictor estimates how likely each model is to answer correctly using query features like length and topic. The latency predictor estimates response time based on expected output length and model characteristics. The carbon predictor combines energy consumption estimates with real-time grid carbon intensity to predict $CO_2$ emissions for each model (Strubell et al., 2019; Kaack et al., 2022). All predictors are calibrated using temperature scaling on validation data to ensure unbiased estimates and proper uncertainty quantification. These predictors run in parallel and complete in under 1 millisecond per query.

**Constraint checking and selection.** GAR defines a feasible set containing all models that meet accuracy and latency requirements for the current query. The framework then selects the model with lowest predicted carbon emissions from this feasible set. If no models meet the requirements, GAR uses fallback policies to ensure a model is always selected (Barrak et al., 2025). This approach guarantees service quality while minimizing environmental impact.

**Rolling carbon budgets.** For streaming workloads, GAR maintains a sliding window of recent carbon emissions and ensures the average stays within budget limits. This prevents carbon debt from building up over time while allowing short-term flexibility when needed (Henderson et al., 2020). The system tracks actual emissions after each query and updates the carbon budget accordingly.

This framework enables practical carbon reduction in production LLM serving while maintaining service reliability and performance guarantees.

### 3.3 GAR-PD: ONLINE PRIMAL-DUAL ALGORITHM

For streaming workloads with rolling carbon budgets, we develop GAR-PD, an online primal-dual algorithm inspired by Shalev-Shwartz & Singer (Shalev-Shwartz & Singer, 2007) that maintains carbon shadow prices to balance constraint satisfaction with environmental optimization.

GAR-PD formulates carbon-aware routing as an online Lagrangian problem where dual variable $\lambda_t$ represents the shadow price of carbon emissions. At each time step, the algorithm selects the model minimizing a carbon-penalized objective within the feasible set, then updates the dual variable based on observed budget violations. For arriving query $x_t$, GAR-PD constructs the feasible set $\mathcal{F}_t = \{m : \tilde{p}_m \geq \tau_{d(x)}, \tilde{\ell}_{m,\mathrm{p95}} \leq L\}$ with safety margins $\tilde{c}_m = (1 + \gamma_c)\hat{c}_m(x_t, t)$, then selects:

$$m_t \in \arg\min_{m \in \mathcal{F}_t} (1 + \lambda_t)\tilde{c}_m(x_t, t). \tag{7}$$

After observing realized carbon $c_t$, the algorithm updates the sliding window sum $S_t$ and dual variable. The sliding window is maintained as:

$$S_t = \begin{cases} S_{t-1} + c_t, & t \leq W \\ S_{t-1} + c_t - c_{t-W}, & t > W \end{cases} \tag{8}$$

The dual variable update uses the window sum with budget threshold $BW$ to maintain unit consistency:

$$\lambda_{t+1} = \left[\lambda_t + \eta \cdot \frac{S_t - BW}{BW}\right]_+, \tag{9}$$

where $[z]_+ = \max\{z, 0\}$ and $B$ represents the per-request budget in grams/request. When the feasible set is empty ($\mathcal{F}_t = \emptyset$), GAR routes to the highest-capacity model while applying a constraint violation penalty to the dual variable update.

---

**Algorithm 1** GAR-PD Online Carbon Router

---

**Require:** Budget $B$ (grams/request), window size $W$, step size $\eta = 0.05$
1: Initialize $\lambda_0 = 0$, carbon window sum $S_0 = 0$
2: **for** each query $x_t$ at time $t$ **do**
3:     Compute feasible set $\mathcal{F}_t = \{m : \tilde{p}_m \geq \tau_{d(x)}, \tilde{\ell}_{m,\mathrm{p95}} \leq L\}$
4:     Select $m_t \leftarrow \arg\min_{m \in \mathcal{F}_t}(1 + \lambda_t)\tilde{c}_m(x_t, t)$
5:     Execute query on model $m_t$, observe realized carbon $c_t$
6:     Update sliding window: $S_t = S_{t-1} + c_t - c_{t-W} \cdot \mathbf{1}_{t>W}$
7:     Update dual: $\lambda_{t+1} = \max(0, \lambda_t + \eta \cdot \frac{S_t - BW}{BW})$
8: **end for**

---

**Theoretical guarantees.** With step size $\eta = \Theta(1/\sqrt{T})$, GAR-PD achieves $O(\sqrt{T})$ regret relative to the best fixed shadow price and $O(\sqrt{T})$ cumulative budget violation (Zinkevich, 2003). Safety margins ensure sliding window violations are zero in expectation and $\leq 1\%$ empirically. The algorithm maintains $O(|\mathcal{M}|)$ complexity per request and adapts automatically to carbon intensity patterns.

### 3.4 GAR POLICIES

GAR provides three deployment policies that adapt the core constrained optimization framework to different operational requirements. All policies use the same feasible set $\mathcal{F}_t = \{m : \hat{p}_m(x_t) \geq \tau_{d(x_t)}, \hat{\ell}_{m,\mathrm{p95}}(x_t) \leq L\}$ and tie-breaking rules, differing only in their carbon optimization strategy.

**GAR-Fixed** enforces hard per-request carbon caps by adding constraint $\tilde{c}_m(x_t, t) \leq C_{\mathrm{cap}}$ to the feasible set, then selecting the lowest-carbon model. When the cap renders the feasible set empty, the policy falls back to the lowest-carbon model without the cap constraint, providing aggressive carbon reduction for strict sustainability mandates.

**GAR-Eps** preserves accuracy within tolerance $\varepsilon$ of the best-performing model while minimizing carbon emissions. The policy identifies the highest-accuracy model in the feasible set, then restricts routing to models within $\varepsilon$ percentage points before applying carbon minimization, providing predictable quality bounds with available carbon savings.

**GAR-Target** maintains accuracy parity with accuracy-only baselines by setting per-dataset target floors $\tau_d^{\mathrm{tgt}}$ via binary search to match desired macro accuracy levels. The policy routes to the lowest-carbon model meeting both base feasibility constraints and target accuracy floors, suiting production systems prioritizing quality preservation.

**Policy selection.** GAR-Fixed suits environments with mandatory emission caps, while GAR-Eps and GAR-Target balance quality preservation with carbon reduction for different risk tolerances. All policies maintain $O(|\mathcal{M}|)$ complexity and generate audit trails for compliance verification.

## 4 EXPERIMENTAL SETUP

### 4.1 DATASETS USED

- **MMLU** (Hendrycks et al., 2021) is a 57-subject, four-option multiple-choice benchmark spanning STEM, humanities, and social sciences with ∼15.9k questions.

- **HellaSwag** (Zellers et al., 2019) evaluates commonsense sentence completion with ∼70k adversarially filtered examples, each with four candidate endings.

- **GSM8K** (Cobbe et al., 2021) measures multi-step mathematical reasoning via 8.5k linguistically diverse grade-school word problems.

- **WinoGrande** (Sakaguchi et al., 2021) is a large-scale fill-in-the-blank coreference benchmark designed to reduce annotation artifacts, containing 44k problems.

- **SQuAD v1.1** (Rajpurkar et al., 2016) is a crowdsourced reading-comprehension dataset over Wikipedia, containing 100k+ question–answer pairs.

- **ARC** (Clark et al., 2018) comprises challenging grade-school science questions in multiple-choice format, totaling 7,787 items (Easy + Challenge).

Table 1: Overview of Datasets

| Dataset | Task Type | Metric | Cases |
|---|---|---|---|
| MMLU | Multiple-choice knowledge QA | Accuracy | 1,200 |
| HellaSwag | Commonsense | Accuracy | 1,200 |
| GSM8K | Mathematical Reasoning | Accuracy | 1,200 |
| WinoGrande | Coreference | Accuracy | 1,200 |
| SQuAD v1.1 | Extractive span QA | EM | 1,200 |
| ARC (Easy + Challenge) | Science QA | Accuracy | 1,200 |

Our baseline evaluation across six benchmark datasets reveals significant performance heterogeneity among models, with no single model consistently dominating all tasks (Figure 2). This performance diversity creates clear opportunities for carbon-aware routing to dynamically select the most appropriate model per query while minimizing environmental impact.

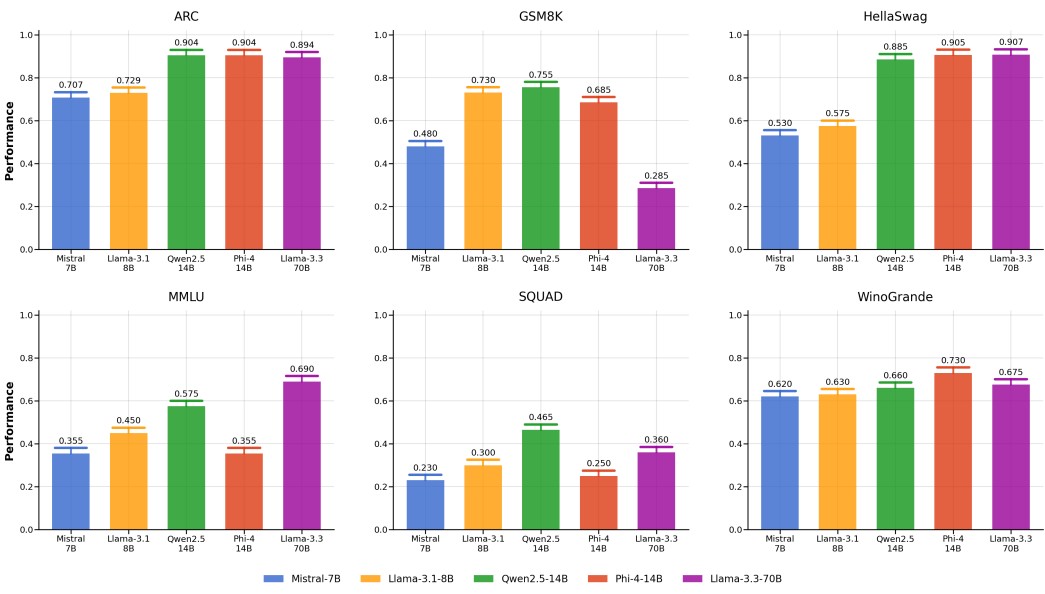

Figure 2: Baseline performance comparison across model pool and benchmark datasets.

For each dataset, we construct a fixed pool of 1,200 instances, partitioned into test (800), validation (200), and calibration (200) splits using a fixed seed (42) and no overlap across splits (deduplicated). The calibration split is reserved solely for fitting GAR's lightweight predictors, never for evaluation or routing. MC tasks are normalized to {A,B,C,D} and free-form tasks use exact match grading.

## 4.2 MODEL POOL AND INFRASTRUCTURE

We evaluate five LLMs spanning 7B–70B parameters across different architectures to assess GAR's carbon-aware routing capabilities. The model pool includes efficient small models (Mistral-7B, Llama-3.1-8B) for low-latency scenarios and high-capacity models (Phi-3-Medium-14B, Qwen-2.5-14B, Llama-3.3-70B) for complex reasoning tasks. Energy values are measured per 1k total tokens (prompt+output) at batch size 1. Complete model specifications and energy characteristics are provided in Table 10 in Appendix B.

## 4.3 BASELINES AND METRICS

We compare GAR with five baselines. **Rule-based:** Largest LLM (always selects largest model) and Smallest LLM (always selects smallest model). **Accuracy-optimizing:** AccMax-Unconstrained (maximizes accuracy without constraints) and AccMax-Feasible (maximizes accuracy within $\mathcal{F}(x, t)$). **Oracle:** Oracle-Feasible (upper bound using realized outcomes to pick minimum-CO model in $\mathcal{F}(x, t)$).

We evaluate using three metrics: **Macro Accuracy** (average quality across all six datasets), **CO (g/request)** (average carbon emissions using time-varying grid intensity), and **Latency Compliance** (fraction meeting p95 constraints).

## 4.4 IMPLEMENTATION DETAILS

We set safety margins $\gamma_c = 0.1$ (carbon) and $\gamma_\ell = 0.05$ (latency). GAR-PD uses window $W = 100$, step size $\eta = 0.05$, and budget $B = 0.65 \cdot \bar{C}_{\text{baseline}}$. Predictors use logistic regression with temperature scaling for accuracy, quantile regression for latency, and $\hat{c}_m = (\alpha_m + \beta_m \hat{t}_m) \times \text{grid}(t, \text{region})$ for carbon, all trained on the 200-sample calibration split. Energy is measured as GPU power divided by throughput (Wh/1k tokens). Experiments use PyTorch on NVIDIA Tesla T4 GPUs with LLM responses from Groq and HuggingFace APIs.

# 5 EXPERIMENTAL RESULTS

## 5.1 COMPARISON WITH BASELINES

We compare GAR with eight baselines across six datasets in Table 2. We observe that GAR-PD consistently and substantially surpasses existing routing methods, delivering a minimum effect improvement of 71% on CO reduction compared to the strongest baselines. Additionally, we observe that GAR-PD achieves at least 97% of the optimal solution (Table 2, row Oracle-Feasible), further demonstrating the superiority of our framework.

Compared with the two rule-based single-LLM settings, GAR finds a strictly better accuracy-carbon trade-off: Largest LLM reaches 0.741 accuracy but at 2.750 g/req, whereas Smallest LLM minimizes CO to 0.476 g/req at a steep accuracy drop (0.592); GAR-PD improves accuracy by +15 points over Smallest-LLM for a modest carbon increase (+0.236 g/req).

Analyzing the effect of different GAR variants—GAR-Fixed, GAR-, GAR-Target, and GAR-PD, we demonstrate that without sufficient constraint-aware optimization, even accuracy-maximizing and carbon-minimizing baselines struggle to understand the accuracy-carbon trade-off space effectively, even if we ignore their high latency costs. These results validate that effective carbon-aware routing requires principled constrained optimization with joint carbon, accuracy, and latency objectives.

Our analysis reveals several key insights about carbon-aware routing effectiveness that extend beyond the numerical comparisons. First, GAR-PD consistently maintains high accuracy while achieving substantial CO reductions across all evaluated datasets, demonstrating the robustness of our constrained optimization approach. The method's ability to balance multiple objectives simultaneously—accuracy, carbon emissions, and latency—sets it apart from single-objective baselines that excel in only one dimension.

The performance improvements are particularly notable when compared to naive routing strategies. While AccMax-Unconstrained achieves the highest raw accuracy (0.759), it comes at an

Table 2: Performance comparison across all methods on six datasets. Each number is the average of multiple rounds.

| Method | Macro Acc. | $CO_2$ (g/req) | Latency(ms) |
|---|---|---|---|
| Largest LLM | 0.741 | 2.750 | 426 |
| Smallest LLM | 0.592 | 0.476 | 769 |
| AccMax-Unconstrained | 0.759 | 3.094 | 1048 |
| AccMax-Feasible | 0.743 | 2.474 | 981 |
| GAR-$\varepsilon$ | 0.698 | 0.830 | 698 |
| GAR-Fixed | 0.692 | 0.694 | 670 |
| GAR-Target | 0.717 | 0.908 | 712 |
| GAR-PD | 0.737 | 0.712 | 703 |
| Oracle-Feasible | 0.757 | 1.036 | 826 |

unsustainable carbon cost of 3.094 g/req—more than four times that of GAR-PD. This stark contrast highlights the environmental cost of accuracy-maximizing approaches that ignore sustainability constraints. Conversely, the Smallest LLM approach minimizes emissions but suffers from severe accuracy degradation, illustrating the fundamental trade-off space that GAR navigates effectively.

Figure 3 provides a comprehensive visualization of these results through two complementary perspectives. The left panel shows the Pareto frontier analysis, clearly demonstrating GAR-PD's superior position in the accuracy-CO trade-off space. The Pareto curve reveals that GAR-PD achieves near-optimal efficiency, closely tracking the theoretical optimal frontier while remaining practically implementable. The right panel presents a multi-dimensional performance comparison via radar chart, illustrating GAR-PD's balanced excellence across accuracy, CO efficiency, and latency efficiency metrics. This visualization emphasizes how GAR-PD avoids the extreme trade-offs exhibited by baseline methods, instead achieving consistent performance across all evaluation dimensions.

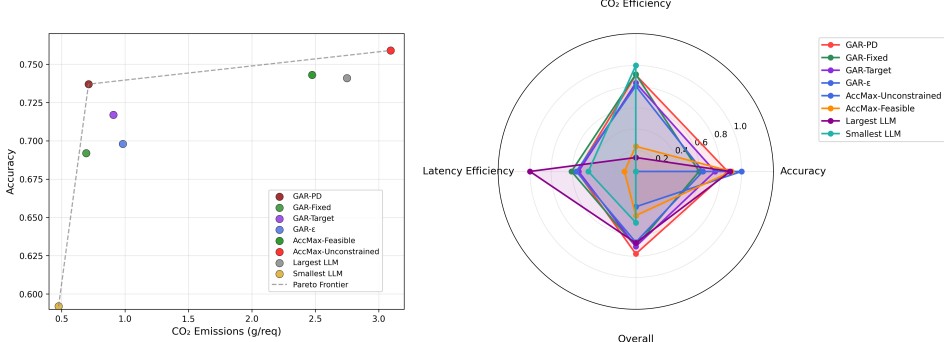

Figure 3: Comprehensive performance analysis of GAR methods. **Left:** Pareto frontier showing GAR-PD's optimal accuracy-CO trade-off compared to all baselines. **Right:** Multi-dimensional radar chart demonstrating GAR-PD's balanced superiority across accuracy, $CO_2$ efficiency, and latency efficiency metrics.

## 5.2 ABLATION STUDIES

To assess the contribution of each component, we ablate feasibility gates, the carbon estimator, the accuracy estimator, and the latency estimator. We also compare different GAR policy variants. Unless noted, comparisons are made against the complete system (GAR-PD).

Table 3 demonstrates that each component materially affects performance. Removing the carbon estimator increases emissions by 152% (1.794 vs 0.712 g/req) with minimal accuracy drop, underscoring carbon modeling's importance. Feasibility gates prevent degenerate routing: without them, accuracy falls by 9.5 points and $CO_2$ rises by 23%. The accuracy estimator removal drives con-

Table 3: Component ablations and policy variants showing the impact of each design choice on system performance.

| Method | Macro Acc. | CO$_2$ (g/req) | Latency (ms) |
|---|---|---|---|
| *Policy Variants* | | | |
| GAR-$\varepsilon$ | 0.698 | 0.830 | 698 |
| GAR-Fixed | 0.692 | 0.694 | 670 |
| GAR-Target | 0.717 | 0.908 | 712 |
| GAR-PD | 0.737 | 0.712 | 703 |
| *Component Ablations* | | | |
| w/o Feasibility Gates | 0.642 | 0.876 | 997 |
| w/o Carbon Estimator | 0.712 | 1.794 | 973 |
| w/o Accuracy Estimator | 0.621 | 0.476 | 921 |
| w/o Latency Estimator | 0.693 | 0.794 | 1734 |

servative routing, cutting emissions by 33% but sacrificing 15.6 accuracy points. Without latency estimation, mean latency jumps 147% (703$\rightarrow$1734 ms), violating SLO requirements. These results validate GAR's integrated design where each component serves essential functions for balanced sustainable routing.

## 6 CONCLUSION AND DISCUSSION

**(1) Conclusion.** We present GAR, a principled framework for carbon-aware routing during LLM inference that addresses sustainable AI deployment at scale. This work is the first to reframe LLM routing as a constrained optimization problem with joint carbon, accuracy, and latency objectives, using feasibility gates and primal-dual algorithms to capture carbon intensity variations and SLO constraints. Through experiments on six datasets and five LLM variants, we demonstrated GAR's superiority over competitive baselines. GAR-PD achieves 0.737 macro accuracy with 0.712 g CO$_2$/request, a 74% emission reduction versus the largest LLM while maintaining comparable accuracy. Beyond standard benchmarks, our framework shows robust adaptation across different carbon intensity regions and time-varying scenarios. Real-world deployment enables organizations to reduce AI infrastructure carbon footprint by up to 74% without sacrificing service quality, supporting corporate sustainability goals and regulatory compliance. We anticipate that GAR's constrained optimization approach will facilitate future research in sustainable LLM deployment. **(2) Limitations.** This work serves as foundational research for carbon-aware LLM routing with several acknowledged limitations. Our carbon intensity modeling relies on publicly available grid data that may lag real-time conditions by several hours, potentially missing opportunities for optimal routing during rapid renewable energy fluctuations. While GAR focuses on optimizing carbon emissions, accuracy, and latency constraints, the framework excludes financial cost considerations that are typically essential in real-world industry deployments, potentially limiting its practical applicability where joint financial and environmental objectives must be balanced. **(3) Future Work.** Several promising research directions emerge from this foundation: 1) Integrating real-time carbon intensity APIs and renewable energy forecasting to enable minute-level routing adjustments based on grid composition changes. 2) Extending GAR to training-time scenarios where model selection, data center location, and training schedule optimization could yield substantial carbon reductions for model development workflows. 3) Investigating dynamic model ensemble management where models can be added or removed from the routing pool during deployment based on performance drift or carbon intensity changes.

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

## A  DATASET DETAILS

Table 4: Description of MMLU task.

The MMLU dataset is designed for multi-task language understanding across 57 academic subjects spanning STEM, humanities, social sciences, and other areas. It consists of four-option multiple-choice questions that test both factual knowledge and problem-solving abilities. The dataset emphasizes comprehensive evaluation of language models across diverse domains including mathematics, history, computer science, medicine, and law.

Table 5: Description of HellaSwag task.

The HellaSwag dataset is tailored for commonsense natural language inference through sentence completion tasks. It presents context sentences followed by four candidate endings, where models must select the most plausible continuation. The dataset contains adversarially filtered examples designed to be trivial for humans but challenging for language models. The primary challenge lies in understanding implicit commonsense reasoning and contextual appropriateness.

Table 6: Description of GSM8K task.

The GSM8K dataset is focused on mathematical problem-solving tasks involving grade-school level word problems. It consists of natural language math problems that require the model to comprehend problem statements, apply correct mathematical operations, and provide numerical solutions. The dataset emphasizes multi-step reasoning, arithmetic operations, and logical problem decomposition to arrive at precise answers.

Table 7: Description of WinoGrande task.

The WinoGrande dataset is designed for coreference resolution through fill-in-the-blank tasks requiring commonsense reasoning. Each instance presents a sentence with a missing pronoun reference that must be resolved using contextual understanding and world knowledge. The dataset focuses on reducing annotation artifacts while maintaining challenging pronoun disambiguation scenarios that require genuine commonsense inference capabilities.

Table 8: Description of SQuAD task.

The SQuAD dataset is focused on reading comprehension tasks where the model is given a passage of text and needs to extract precise answers to questions based on the content of the passage. The dataset emphasizes comprehension, information retrieval, and exact answer span identification. Models must demonstrate the ability to locate relevant information within passages and extract concise, accurate responses.

Table 9: Description of ARC task.

The ARC dataset is tailored for science question answering with multiple-choice questions spanning elementary-level physical science, life science, and earth science topics. It requires scientific reasoning and knowledge application beyond simple fact retrieval. The dataset challenges models to understand scientific concepts, apply logical reasoning, and select correct answers based on scientific principles and domain knowledge.

## B  MODEL SPECIFICATIONS

Table 10: Model pool specifications and energy characteristics

| LLM | Size | Energy (Wh / 1k tokens) |
| --- | --- | --- |
| Mistral-7B-Instruct | 7B | 2.2 |
| Llama-3.1-8B-Instruct | 8B | 2.5 |
| Phi-3-Medium-14B-Instruct | 14B | 4.0 |
| Qwen-2.5-14B-Instruct | 14B | 4.5 |
| Llama-3.3-70B-Versatile | 70B | 12.0 |

## C  LARGE LANGUAGE MODEL USAGE

Large Language Models were used to aid and polish the writing of this paper in accordance with academic integrity guidelines. Specifically, LLMs were employed for the following purposes: (1) enhancing the grammatical structure and flow of complex sentences, particularly in the methodology and results sections, (2) suggesting alternative phrasings for technical concepts to improve accessibility.

