# OpenReview forum: "GAR: Carbon-Aware Routing for LLM Inference via Constrained Optimization"
_ICLR.cc/2026/Conference — ICLR 2026 Conference Withdrawn Submission_

### Official Review · Reviewer_pq4d · 2025-10-15

**Soundness:** 2
**Presentation:** 2
**Contribution:** 2
**Rating:** 2
**Confidence:** 5

**Summary:**

This paper introduces GAR (Green-Aware Routing), which formulates carbon-aware LLM routing as a constrained multi-objective optimization problem that minimizes per-request $CO_2$ emissions subject to accuracy floors and latency constraints. The key methodological contribution is GAR-PD, a partially online primal-dual algorithm that maintains carbon shadow prices through dual variable updates and achieves good regret bounds, using lightweight estimators for correctness, latency, and carbon emissions to enable real-time routing decisions without additional inference passes.

**Strengths:**

- Energy efficiency in LLM inference applications is a key challenge in practice. The paper offers an interesting approach to uniting model performance and cost optimization.
- The problem considers performance, cost, and latency at once, which is a rare and very relevant combination to find in related work

**Weaknesses:**

- I see a fundamental issue in using pre-trained predictors to estimate the carbon emissions for three reasons. First, the energy mix is quite different across the globe and curating individual datasets for local settings is likely impractical. Second, energy markets are very dynamic. Take Europe as an example. Depending on the momentary demand, energy can be imported from other EU countries on the fly, changing the enegy mix and therefore the $CO_2$ emissions quite drastically. Third, and most important, users primarily care about the price they pay. To stay with the EU example, carbon emissions are priced per metric ton and immediately applied on top of the energy price. From a practical perspective, that is likely the only way to really keep track of how much $CO_2$ has been emitted for any generated kWh of energy. In my eyes, the GAR approach would benefit from looking at the operating cost rather than $CO_2$ emissions that are only one part of the operating costs for an LLM or a set of LLMs.
- The paper claims to be the first to present a joint optimization objective for model performance and operating cost/carbon emissions but does not look into directly related work [1, 2], let alone compare against the papers.
- The authors do not analyze the routing overhead (e.g., how much added emissions do I incur for querying 3 predictors?) and it is also not part of the optimization problem. For a complete and thorough analysis, I would hope to see the total emissions of the system.
- The proofs for the theoretical guarantees (L. 238) are missing.


[1] Causal LLM Routing: End-to-End Regret Minimization from Observational Data, Tsiourvas et al., May 2025.

[2] MESS+: Dynamically Learned Inference-Time LLM Routing in Model Zoos with Service Level Guarantees, Woisetschläger et al., May 2025.

**Questions:**

- How are the optimization constraints weighted? At the moment it appears that it is a sequential selection process ("feasible set" in Alg. 1) but does not leave the user with a choice of how to prioritize based on remaining $CO_2$ budget.
- Can you please elaborate more on the safety margins? They appear to be quite large (see Sec. 4.4)
- Can you please provide details on how you measured the energy consumption? Did you use the theoretical TDP of the T4 GPU or did you actually measure the power consumption?
- What are the key differences between GAR and related papers [1, 2]? (see weaknesses)

**Minor remark:**
- L. 340: CO should be $CO_2$.

---

### Official Review · Reviewer_NrWK · 2025-10-22

**Soundness:** 2
**Presentation:** 2
**Contribution:** 1
**Rating:** 0
**Confidence:** 4

**Summary:**

This paper introduces Green-Aware Routing (GAR), a framework for routing LLM inference queries to minimize $CO_2$ emissions while adhering to service-level objectives (SLOs). The authors identify that existing routing methods optimize for cost, latency, and accuracy, but overlook environmental impact, which can vary significantly based on model choice and real-time grid carbon intensity.

The paper formulates this problem as a constrained multi-objective optimization problem, aiming to minimize per-request $CO_2$ subject to explicit accuracy floors and p95-latency guarantees. The framework relies on lightweight pre-route estimators for correctness, latency, and carbon emissions to make real-time decisions. The primary algorithmic contribution is GAR-PD, an online primal-dual algorithm designed to manage a rolling carbon budget by updating a carbon "shadow price". The paper also proposes several practical heuristic variants (e.g., GAR-Fixed, GAR-Target).

Experiments conducted across six standard NLP benchmarks  and a pool of five LLMs (7B to 70B)  demonstrate that the proposed methods can achieve substantial carbon reductions.

**Strengths:**

1. **Principled Formulation:** The decision to formulate the problem as a *constrained* optimization problem (minimizing $CO_2$ *subject to* accuracy and latency SLOs) makes sense. It correctly reflects production requirements where service quality is a hard constraint, rather than just another factor in a simple trade-off.
2. **Practicality:** The use of lightweight, pre-route estimators for quality, latency, and carbon makes the framework practical for real-world deployment, as these checks are fast (reportedly < 1ms).

**Weaknesses:**

1.  **Foundational Flaw in GAR-PD Algorithm:** The paper's main algorithmic contribution, GAR-PD, appears to be fundamentally flawed. The selection rule is defined in Equation 6 as $m_t \in \arg \min_{m \in \mathcal{F}(x,t)} (1+\lambda_t)\tilde{c}_m(x,t)$. At any given time $t$, the term $(1+\lambda_t)$ is a non-negative scalar that is identical for all models $m$ in the feasible set $\mathcal{F}(x,t)$. Therefore, multiplying the carbon cost $\tilde{c}_m$ by this scalar does not change the $\arg \min$. The selection rule for GAR-PD is mathematically identical to the selection rule for the basic GAR policy in Equation 5, $R^{GAR} \in \arg \min \tilde{c}_m(x,t)$.
2.  **Missing Theoretical Guarantees:** The paper claims that GAR-PD achieves $O(\sqrt{T})$ regret and cumulative budget violation, citing standard online learning literature. However, it provides no formal theorem statement, no discussion of the assumptions required, and no proof sketch showing how those results (which typically apply to online convex optimization) are adapted to this specific problem of discrete, constrained model selection with a rolling budget window.
3.  **Ambiguous Experimental Reporting:** The core constraint is on *p95-latency* (Eq 3: $\tilde{l}_{m,p95}(x) \le L$ ). However, Table 2 reports a single, ambiguous "Latency(ms)" metric. It is not stated whether this value represents the average (p50) latency or the p95 latency. If it is the average, the results fail to demonstrate that the p95-latency SLO is actually being met.
4.  **Unfulfilled Robustness Analysis:** The abstract and introduction explicitly promise "comprehensive robustness analysis under prediction miscalibration and distribution shift." This analysis is not present in the paper. Section 5.2  is an ablation study, not an analysis of the system's robustness to, for example, a systematically over- or under-confident accuracy estimator or a miscalibrated carbon predictor.

**Questions:**

See the weaknesses. The authors should supplement all the missing parts for a qualified paper.

---

### Official Review · Reviewer_X118 · 2025-11-02

**Soundness:** 2
**Presentation:** 3
**Contribution:** 2
**Rating:** 4
**Confidence:** 3

**Summary:**

This paper proposes a green-aware routing (GAR), which aims to minimize per-request carbon emission by intelligently routing requests to appropriate LLMs while meeting the accuracy and latency targets. GAR uses small predictors for carbon emission, response quality, and latency along with an online algorithm to make the routing decision. The experimental results suggest that GAR can significantly reduce the carbon emission while maintaining the average accuracy and latency.

**Strengths:**

The sustainability of large-scale LLM inference is an important challenge to address. This paper seems to represent one of the first work on studying how routing of individual requests can be optimized to reduce carbon emission.

The experimental results suggest that the proposed approach has the potential to significantly reduce the carbon emission compared to naïve strategies.

**Weaknesses:**

The paper presents the proposed predictors and the experimental results at a relatively high level and only compares GAR with relatively naïve routing policies. In order to more fully understand the technical novelty/contributions and the impact of GAR on accuracy/latency/carbon, the paper needs to provide more details.

1) Comparison with other LLM routing schemes
To show GAR's advantages over the state-of-the-art, the paper needs to compare GAR with other LLM routing schemes previously proposed. Even if previous schemes may not have explicitly optimized carbon emission, optimizing energy consumption or LLM resource usage can also help carbon emission. In that sense, it will be good to see how GAR can reduce carbon emission compared to other predictive LLM routing schemes. Also, it will be helpful if the paper shows the performance of previous schemes when a carbon predictor is added. This comparison will help clarifying whether the main contribution comes from simply including carbon emission as an optimization target or more from the optimization algorithm (PD).

2) Detailed breakdown of results
The paper only presents the average accuracy/latency across all benchmarks and queries. Even when the average accuracy/latency is maintained, dynamic optimization scheme may negatively impact a subset of tasks or queries. In that sense, the paper needs to show how GAR affects the accuracy/latency for individual benchmarks and queries.

3) Details of the predictors
The paper only describes the predictors briefly in Section 4.4. Given that the accuracy of lightweight predictors will directly impact the optimization performance, I would suggest providing more details of the predictors and clearly point out the new/novel aspects of the design. Also, it will be important to understand exactly which datasets are used to train (and test) the predictors. For broadly applying GAR, it will be important to show that the predictors generalize beyond the training datasets.

4) Other minor points
For practical deployment, it will be good to understand the performance overhead of the proposed lightweight predictors and the online routing algorithm. Also, some of the experimental results need more explanation. For example, it is not clear why 'Smallest LLM' leads to higher latency compared to 'Largest LLM' in Table 2. Intuitively, it seems small LLMs should run faster compared to large LLMs.

**Questions:**

See the questions in the weakness section.

---

### Official Review · Reviewer_zyHG · 2025-11-11

**Soundness:** 3
**Presentation:** 3
**Contribution:** 2
**Rating:** 4
**Confidence:** 3

**Summary:**

The authors present a framework, Green-Aware Routing (GAR), for carbon-aware routing during LLM inference with heterogeneous workloads. Framed as an online constraint optimization problem, a given request arriving at a given time is assigned to a particular model to fit conditions for latency, accuracy(/task performance), and carbon emissions. Comparing to basic baselines and through ablations, the authors find that specifically predicting carbon emissions and not relying solely on cost or performance is a critical component of their framework.

**Strengths:**

1. Creative and theoretically grounded framework for reducing carbon emissions associated with LLM inference and deployment while maintaining adherence to latency SLOs and quality standards
2. Well-written introduction and background sections
3. Thoughtfully scoped experimental setting with multiple tasks, a model pool, and a number of baseline comparisons and ablations

**Weaknesses:**

1. Some of the more consequential aspects of the experimental methodology are not clearly explained. Perhaps I simply missed it, but I did check again in what would seem to be reasonable locations. In particular, what exactly is run in the experiments? What does the distribution of arrival times look like in the models?
2. It could be argued that the datasets chosen do not capture a realistic serving case that would call for routing between LLMs -- why not use a trace? (Relatedly, **missing a critical related work https://arxiv.org/abs/2408.00741** -- chat and code traces are released with the DynamoLLM paper)
3. Weak baselines. +See Q4 below: it is suggested that purely optimizing for accuracy or any other single objective is not enough, but afaik there are not ablations that answer the question of how beneficial carbon-aware routing specifically is when directly compared to routing methods that focus on e.g. latency which is to my understanding highly correlated with carbon emissions

**Questions:**

1. Who exactly is the intended user of this framework? An individual practitioner? a provider? In particular, what assumptions are relevant here? (Important to know because there would probably be different considerations if we could assume all models had dedicated permanent deployments vs sometimes needed to be spun up or down depending on usage)
2. See weakness #1 above
3. What exactly are the overhead costs associated with training and running the “lightweight predictors trained on historical data” mentioned?
4. What happens if, instead of ablating the carbon estimator altogether, latency was used as a proxy for carbon emissions?

Minor clarification questions:
1. In table 10 are the 1k tokens input or output tokens?
2. What exactly is W in equation 8?
3. More of a suggestion than a question, but it would be nice to see more details on the datasets that are relevant to inference efficiency (i.e.

---

### Note · Authors · 2025-11-22

I have read and agree with the venue's withdrawal policy on behalf of myself and my co-authors.